# Sports and Energy Drink Consumption, Oral Health Problems and Performance Impact among Elite Athletes

**DOI:** 10.3390/nu14235089

**Published:** 2022-11-30

**Authors:** Kamran Khan, Abdul Qadir, Gina Trakman, Tariq Aziz, Maria Ishaq Khattak, Ghulam Nabi, Metab Alharbi, Abdulrahman Alshammari, Muhammad Shahzad

**Affiliations:** 1Oral and Maxillofacial Surgery Unit, MTI Hayat Abad Medical Complex, Hayat Abad Phase 4, Peshawar 25100, Pakistan; 2Institute of Basic Medical Sciences, Khyber Medical University, Peshawar 25120, Pakistan; 3Department of Dietetics, Nutrition and Sport, La Trobe University, Melbourne 3086, Australia; 4School of Food & Biological Engineering, Jiangsu University, Zhenjiang 212013, China; 5Institute of Public Health and Social Sciences, Khyber Medical University, Peshawar 25120, Pakistan; 6Institute of Nature Conservation, Polish Academy of Sciences, 71-330 Krakow, Poland; 7Department of Pharmacology and Toxicology, College of Pharmacy, King Saud University, P.O. Box 2455, Riyadh 11451, Saudi Arabia; 8School of Biological Sciences, Health and Life Sciences Building, University of Reading, Reading RG6 6AX, UK

**Keywords:** sports, athletes, self-reported oral health, gut microbiota, caries, periodontitis

## Abstract

Frequent consumption of sports and energy drinks among athletes is of concern due to its detrimental impact on oral health. The present study aimed to assess sports and energy drink consumption, oral health status and impacts on daily activities and sports performance among elite athletes from Pakistan. Data regarding socio-demographic characteristics, sports and energy drink consumption, oral health and hygiene practices, self-reported oral health and psychosocial and performance impact was assessed using a self-administered, structured questionnaire followed by clinical oral examination by a single, experienced dentist. A total of 104 athletes, a majority of whom were male (80.8%), participated in the study. Around two third of the participants reported consumption of sports and energy drinks, energy gels or bars at least once a week, the commonest being Sting. Despite good oral hygiene practices, the athletes generally had poor oral health with high prevalence of dental caries (63.5%), gingivitis (46.1%), irreversible periodontitis (26.9%) and erosive tooth wear (21.2%). More than a quarter (28.8%) of the athletes rated their oral health as fair—very poor. Four in five athletes (80%) also experienced at least one oral problem with negative impacts on daily activities (64.4%) and participation training and sports performance (36.5%). Regression analyses revealed a significant association between periodontal disease and impact on both daily activities and sports performance. To our knowledge, this is the first study reporting that high-prevalence sports and energy drink consumption and oral problems among elite athletes from Pakistan has a negative impact on daily activities and sports performance. These findings may have important implications for oral health education programs, and the need to create awareness among the athletes regarding the use of sports and energy drinks, as well as regular oral health screening of athletes to minimize the impact on performance.

## 1. Introduction

Sports and energy drinks are non-carbonated, sugar-sweetened beverages that are believed to prevent or reduce fatigue, help replenish the body during exercise, and improve cognitive and physical performance and are thus very popular among adolescent and athletes across the world [1]. Although the composition varies widely across the type and brands, the majority of these beverages contain an enormous amount of sugar (up to 10%) [2]. The presence of high amounts of sugar in soft drinks, including sports and energy drinks, has been raising serious concerns among healthcare professionals regarding detrimental health consequences. For examples, a recent population-based cohort study involving 451,743 participants from 10 European countries reported a significant association between sugar-sweetened beverages and all-cause mortality [3]. Although, the mechanism underlying increased mortality is not known, emerging research evidence suggest that frequent and high intake of sugar-sweetened beverages alters gut microbiome composition and functional potential (dysbiosis) [4], thus contributing towards obesity, cardiovascular diseases and increased mortality [5]. Gut microbial dysbiosis associated with sugar-based beverages has frequently been reported in both animal studies and human-based clinical trials [6]. In addition to systemic health and mortality, consumption of sugar-based soft drinks including sports and energy drinks have been shown to negatively affect oral health. In this context, several studies have reported high prevalence of oral problems such as dental carries and erosion associated with increased consumption of energy and sports drinks, especially among children, adolescents and athletes [7,8,9]. Furthermore, research evidence also suggests the negative impact of oral health problems on athlete’ performance and wellbeing [10].

Sound oral health and dentition is an important component of one’s general health and a core enabler of psychosocial participation. However, it is traditionally ignored by athletes and sports professionals. Athletes often tend to focus more on maintaining and/or improving physical fitness and health [11] while ignoring other aspects of health, such as oral health and preventive dental care [12]. Since elite athletes, especially those in contact sports, are at high risk of injuries, research on oral health among athletes has traditionally been limited to oral trauma and use of the mouthguards [10]. However, recently, oral diseases and conditions have been shown to effect physical fitness and athletic performance [10,13]. In the last two decades, sports dentistry has emerged as a specialized field within dentistry that aims to prevent and manage oro-facial trauma and associated oral diseases particularly those affecting performance in athletes.

Athletes are generally at high risk of poor oral health for several reasons. To achieve peak performance, athletes in competitive sports frequently consume foods, beverages, sports drinks, energy bars and gels that are rich in carbohydrates and energy [14]. Even though the low pH and acidic nature of energy and sports drinks favor tooth erosion, these products are usually marketed without expert guidelines regarding oral health. The problem is further complicated in endurance athletes because intense exercise can lead to immunosuppression, which alters saliva composition and therefore further increases risk of poor oral health [15].

The high prevalence of untreated oral problems in athletes was first reported at the 1958 FIFA World Cup in Sweden, during which a dentist performed 118 tooth extraction surgeries on 33 athletes from various countries [16]. More recently, research conducted during top-level international sports competitions has shown similar trends. For example, during the 2004 Olympic Games at Athens, dental care was the second most common requested healthcare service by athletes [17]. During the 2008 Beijing Olympic Games, as many as 1600 dental treatments were carried out [18]. Likewise, high prevalence’s of dental caries (55%), gingivitis (76%) and periodontitis (14%) were found in athletes participating in the London 2012 Olympic Games, and as a result, 30% of the medical emergencies presented during the games were related to oral and dental problems [19,20]. More recently, oral health screening of the Dutch elite athletes before the 2016 Rio Olympic Games revealed that almost 50% of athletes required dental treatment to ensure healthy participation in the games [21].

There exist several mechanisms by which oral problems can exert a negative impact on athletic performance. Dental malocclusion [22] and bruxism [23] not only affect postural position and stability of the mandible but are also associated with pain in the teeth (due to tooth wear) and muscles of the head, neck, back and joints (temporomandibular joints). As a result, rest and muscle repair during sleep [16,24] is greatly impaired in athletes with bruxism, which may have flow-on negative impacts on sport performance. Similarly, dental caries, periodontal diseases and periapical infections can also act as infectious foci for local oral tissues. If left untreated, these conditions can elicit a systemic inflammatory response [25] and therefore may affect athletes’ physical fitness, performance, and wellbeing [26,27,28].

Although recent research suggests athletes are at risk of poor oral health, and that oral health can negatively impact athletic performance, to our knowledge, no studies have evaluated oral health, associated factors and performance impact among elite Pakistani athletes. Therefore, the aim of the current study was to evaluate sports and energy drink consumption, oral health and hygiene practices, and self-reported oral health problems and their impact on daily activities and sports performance among elite Pakistani athletes, to create awareness and influence policymaking in this regard.

## 2. Materials and Methods

### 2.1. Study Design and Population

This was a cross-sectional study conducted among elite athletes from Peshawar, the capital city of Khyber Pakhtunkhwa, a province of Pakistan. Adolescent or adult athletes who competed in any sports at college/university, at a national or international level, and were training at least 10 h per week were eligible to participate in the study. Non-consenting individuals and those who played sports for recreational purposes were excluded from the study. Participants were recruited from Qayyum Stadium and Hayat Abad Sports Complex, the two main sporting arenas in Peshawar. The researchers visited the sites during the training hours; eligible participants were provided with an information sheet containing detailed information about the study and procedure in the national language (Urdu) and were given an opportunity to ask questions related to the study before signing written informed consent if they were interested in participating. For participants below 18 years of age, informed consent was signed by their parents after assent from child. The athletes were required to make an appointment with a designated dental clinic for free oral-health screening within three months of signing the informed consent.

### 2.2. Data Collection

Data regarding sports and energy drink consumption patterns (frequency and time of consumption), brand name and the reason athletes consume these drinks was collected using a previously published questionnaire with only slight modifications [29]. Socio-demographic, oral health behaviors, and psychosocial and performance impact data were collected using a paper-based, self-administered, structured questionnaire developed locally with input from experts in academia and research, sports professionals and dentists (n = 6). Before implementation, the questionnaire was pre-tested on 10 elite athletes who were not part of the main study. The final questionnaire had 34 open-ended and multiple-choice questions divided into four sections. Section A included questions on socio-demographic characteristics such as age, gender, household income, athletic caliber, etc. Section B included information on the oral health behaviors and practices of the athletes. Section C was adapted from a self-reported oral health questionnaire [30], a simple and comprehensive tool for assessing perceived oral health status and satisfaction. Section D included questions about the impacts of oral health on the athletes’ psychosocial life and performance [13]. Since the education system in Pakistan is in English, and the national language is Urdu, the questionnaire contained items in English with Urdu translation in the same box. The questionnaire was provided to the participants at the time of recruitment with instructions to fill it out at home, at a convenient time, and return to the research team when visiting the dental clinic for oral health screening. All questionnaires are included as Appendix A.

### 2.3. Clinical Oral Examination

Clinical oral examinations of all the participants were conducted in an easily accessible and well-equipped private dental clinic. To avoid inter-individual variations in clinical assessment, and to acquire uniformity in the clinical parameters, all the examinations were conducted by a single, experienced clinical dentist. The participants were examined supine in a dental chair with illumination from a mobile lamp above the chair. Prior to examination, compressed air from a portable dental unit was used to dry the teeth. For each athlete, a separate sterile set of dental examination instruments was used. A DMFT (decayed, missing, filled teeth) score was used to record the presence of dental caries (enamel or dentine), missing teeth and restorations [31]. The presence of dental caries, literally known as tooth decay, was assessed using a metallic periodontal probe. Decay was confirmed in the presence of a carious lesion with an occlusal or smooth surface, an unmistakable cavity, undermined enamel, or a detectably softened floor or wall. Periodontal health assessment was done using Basic Periodontal Examination (BPE) [32]. BPE is a simple and rapid screening tool to assess periodontal health and treatment need in community settings and dental screening. BPE was performed by dividing the mouth into six sextants and examining periodontal tissue around all the teeth (except third molars) with a WHO probe (Community Index of Periodontal Treatment Need probe) using low force. The highest score was recorded for each sextant. BPE score range from 0 to 4, where 0 = healthy gingiva with no bleeding on probing, 1 = gum bleeding on gentle probing, 2 = supra or sub-gingival calculus or restorations overhangs, 3 = presence of a periodontal pocket (4.5 mm), 4 = periodontal pocket > 6 mm. Erosive tooth wear assessment was carried out using Basic Erosive Wear Examination (BEWE) [33]. A BEWE score uses a four-point scoring (0–3) system for grading erosive tooth wear. The highest score in each sextant is recorded and a cumulative score is calculated for all sextants. A BEWE score > 7 is usually indicative of non-physiological tooth wear.

### 2.4. Data Analysis

All the data collected was entered into Microsoft Excel 2013 and then exported into SPSS Version 23 (IBM Corp., Armonk, NY, USA) for data analysis. Normality of the data was assessed by visual analysis of histograms. Questionnaire responses regarding socio-demographic characteristics, oral health behaviors, and sports and energy drink consumption underwent a frequency or descriptive analysis; categorical variables are reported as N, % and continuous variables are reported as mean ± standard deviation, or median and interquartile range, as appropriate.

Differences in BEWE, BPE and DMFT scores were all non-normally distributed, and differences in these measures based on socio-demographics; teeth-brushing habits; and reported energy/sports drink, bar, and gel consumption (yes/no) were assessed by Mann-Whitney U-test or Kruskal–Wallis test. Differences in self-reported oral health based on socio-demographics; teeth-brushing habits; and reported energy/sports drink, bar, and gel consumption (yes/no) were assessed by chi-square analyses; for multinomial variables, adjusted residuals were evaluated when the chi-square test was statistically significant (*p* < 0.05) to evaluate which values differed from expected values.

Clinical indicators of oral health were dichotomized as follows: erosion (BPE 0 vs. BPE ≥ 1), periodontal conditions (BPE 0 vs. BPE ≥ 1), and presence of dental caries (DMFT vs. DMFT ≥ 1). The impact of these factors and self-reported oral health (at least one problem vs. no problems) on daily activities (no impact vs. at least daily activity impacted) and participation in sports, training and performance (yes/no impact) were assessed using crude and adjusted logistic regression and were dichotomized.

## 3. Results

### 3.1. Background Information and Characteristics of the Athletes

Of 125 eligible athletes invited to participate in the study, 104 athletes were able to undergo oral health screening, hence the overall response rate was 83.2%. The background information of the athletes is summarized in Table 1. Overall, the mean age of the participants was 18.5 years and the majority were males (80.8%). Participants represented nine sports categories, with table tennis and athletics (track and field) being the most common. Around two-thirds of the athletes played sports at the national and international levels. On average, the participants spent 26 ± 10.4 h per week in training. A majority (46.2%) of the athletes reported studying or having studied at college or university.

### 3.2. Oral Hygiene Habits of the Athletes

Participants were asked about their oral hygiene habits. As shown in Table 2, almost all (98.1%) of the participants reported that they clean their teeth regularly at least once a day (49%). A majority (86.5%) of participants used a toothbrush (86.5%) or miswak (11.5%) to clean their teeth. Dental floss was used by only 2 (1.9%) participants. In terms of dental-care access, around 40% of the athletes reported that they had never visited a dentist and only 5% visited a dentist regularly.

### 3.3. Sports and Energy Drink consumption

Two-thirds of participants (66.7%) reported use of at least one energy drink, bar or gel per week. As shown in Table 3, the majority of participants (n = 33) reported that they usually drank Sting. Consumption of other brands, such as Red bull and Booster, was also reported. These products were commonly used before, during and after training and less commonly used during competitions. The most often-cited reason for consumption of sports and energy drinks among athletes was the belief that these products provide energy, reduce fatigue and stress, and improve performance.

### 3.4. Self-Reported Oral Health and Impact of Demographics, Oral Hygiene Habits and Energy Drink Consumption on Oral Health Status

Fourteen (13.5%) participants rated their general and oral health as fair, and 30 (28.8%) rated it as very poor. The majority of participants (83, 79.8%) also reported experiencing at least one problem related to oral or dental health in past year, with the commonest being dental pain, reported by 33 participants (31.7%), and sensitivity to hot or cold, reported by 27 participants (26%). Additional data on self-reported oral health is outlined in Table 4. A higher proportion of young participants (*p* = 0.011) and married participants (*p* = 0.005) self-reported at least one oral health problem. Likewise, self-reported oral health problems differed across education status (*p* = 0.005), with those with college/university-level education differing from expected values (adjusted residual −3.1 to 3.1). Mother tongue; sport played; level of education; nutrition education; income; teeth-brushing habits; and consumption of sports/energy drinks, bars and gels did not impact self-reported oral health.

### 3.5. Oral Health Impact on Daily Activities and Performance

Sixty-seven (64.4%) participants reported difficulty with at least one daily activity because of problems in their mouth, teeth or gums. Nearly half (47.1%) reported that they had difficulty eating or drinking, with 42 (40.4%), 37 (35.5%), and 38 (36.5%) reporting that oral health problems led to difficulty in relaxing, participating in sports, and smiling without embarrassment, respectively (Figure 1).

### 3.6. Impact of Demographics, Oral Hygiene Habits and Energy Drink Consumption on Oral Health Status

Of all the participants, 66 (63.5%) had at least one carious tooth as assessed by DMFT score. The mean DMFT score was 2.7 ± 3.3. The DMFT score was higher in males compared to females (2.0 vs. 0.5, *p* = 0.013) and differed across sport played (*p* = 0.039).

We found excellent periodontal health (BPE score = 0) in only 28 (26.9%) of the participants. Nearly half the participants (48, 46.1%) presented with clinical gingivitis (BPE score = 1–2), while irreversible periodontitis (BPE score = 3–4) was present in more than a quarter of the athletes. BPE differed across sport played (*p* = 0.022) and income categories (*p* = 0.016). BPE was also higher in those consuming sports/energy drinks (1.5 vs. 2.0, *p* = 0.037) and was lower in those who stated they never consumed any sports/drinks or bars or gels (1.0 vs. 2.0, *p* = 0.008), but no differences were found when sports bars and gels were evaluated independently.

Erosive tooth wear was also common and reported by 22 participants. BEWE (differed across sports played (*p* = 0.039) and mother tongue (*p* = 0.002) but there were no other statistically significant differences in the normality on demographics and consumption of sports/energy drinks, gels, or bars. Likewise, there were no differences for any clinical measures or oral health (DMFT, BPE, BEWE) based on age, marital status, athletic caliber, level of education or nutrition education. There were also no differences based on teeth-brushing (yes/no) and frequency of teeth brushing (1/day vs. 2–3/day), but only two respondents reported they never brushed their teeth.

### 3.7. Factors Affecting Daily Activities and Sports Performance in Athletes

For impact on daily activities, the crude logistic regression model was statistically significant for BPE only, χ2(1) = 10.563, *p* = 0.004. The model explained 1.5% (Nagelkerke R2) of the variance in participants and correctly classified 64.4% of cases. For impact on performance participation in sports, the crude logistic regression model was statistically significant for BPE only, χ2(1) = 6.227, *p* = 0.013. The model explained 8.0% (Nagelkerke R2) of the variance in participants and correctly classified 63.5% of cases. These results are further summarized in Table 5.

Adjusted models were run to correct for variables that were found to have a statistically significant impact on BPE (sport played; income; sports drink consumption; sports/energy drink, bar or gel consumption). For impact on daily activities, the adjusted logistic regression model was statistically significant, χ2(5) = 14.402, *p* = 0.013. BPE remained significant (0.163 [0.042, 0.628], *p* = 0.008), and was the only significant variable in the model. The model explained 54.1% (Nagelkerke R2) of the variance in participants and correctly classified 47.4% of cases.

For impact on participation in sports, the adjusted logistic regression model was not statistically significant (χ2(5) = 9.735, *p* = 0.083); however, BPE remained a significant predictor of impact on participation in sport (0.154 [0.029, 0.818], *p* = 0.028).

## 4. Discussion

Sound health, nutrition and training are important factors for athletes’ health, wellbeing and optimum performance in sports. Neglecting any aspect that affects athlete health and wellbeing can lead to sub-optimal performance and failure in sports competitions [10]. The current study aimed to survey sports and energy drink consumption, assess oral health status, and identify any impacts on daily activities and sports performance among elite athletes from Pakistan. The study provides preliminary evidence about the prevalence or sports and energy drink consumption, oral health problems and the self-reported impacts of these issues.

Overall, the current study reported a high prevalence (66.7%) of sports and energy drink consumption among elite athletes. Our study findings are similar to previous studies which found that 64.9% [29] and 62.2% [34] of participants consumed sports and energy drinks. Other studies have reported an even higher prevalence of energy drink consumption among university athletes: 73% [35] and 86.7% [36]. Higher consumption of sports and energy drinks among athletes is not surprising since many advertisement campaigns link these products to sports and performance [37], making them popular among athletes and the young, physically active population. Athletes further elaborated that they use these products because they provide them with energy and help reduce fatigue and stress. Our study findings are in concordance with previous reports [29,37], thus supporting Bonci’s [38] suggestion that people use energy drinks because they believe these products will help them obtain extra energy in a relatively easy and quick way, thus helping them undertake vigorous activities and recovery from exercise fatigue.

While energy and sports drinks can be beneficial for athletes, in many sporting events plain water is an appropriate beverage of choice [3,5]. Given the high prevalence of sports drink consumption and athletes’ stated reasons for choosing these drinks, it is prudent to provide athletes with education regarding the ingredients and possible side-effects, especially those associated with overdosing and/or prolonged use. For examples, a closer look at the ingredients of the most commonly consumed energy drink (Sting™) reveals the presence of a high amount of sugar (34.3 mg) and caffeine (200 mg) per 250 mL serving. The contents are markedly higher than the FDA recommended daily intake and regular use of this drink may have several detrimental effects on health. Furthermore, it is also crucial for athletes to have knowledge about the types of sporting events where sports drinks are beneficial, as well as education regarding the detrimental consequences of sports drinks on health [31], including oral health. Indeed, the clinical oral examination of the athletes revealed several oral health issues.

We reported dental caries as the most common oral problem, affecting 63.5% of the elite athletes surveyed. Notably, previous studies among athletes in different sports categories [19,23,39,40,41,42,43] have reported lower prevalence of dental caries, ranging from 36.0% in professional football players in the United Kingdom [44] to 55.1% of Olympic athletes from Africa, Europe and the America’s [23]. The higher prevalence of caries in our participant group may be explained by variance in geographic location, as existing studies were predominately conducted in developed nations, which could influence access to dental care and whether or not the water supply is fluoridated [23,43].

Theoretically, frequent carbohydrate intake in the form of sports drinks and beverages [15,44] may enhance the growth of salivary cariogenic bacteria (such as *S. mutans, Lactobacillus* spp.) and decrease salivary Ig-A levels [45], thus making the athletes prone to develop dental caries. While our study did not have a non-athlete comparator group, the prevalence of dental caries in athletes and non-athletes appears to be comparable, with a recent meta-analysis of 30 studies encompassing 27,878 subjects, reporting that nearly 60% of the general population in Pakistan have dental caries [46]. This is in contrast to previous studies, which have reported he prevalence of caries is higher among active soccer players than inactive controls [45,47,48]. The lack of difference in rates of dental caries between athletes and non-athletes in our study may be related to the relatively poor diet quality of the general population in Pakistan, as well as other factors, such as smoking, which can influence oral health.

Regarding periodontal health, we observed a high prevalence of gingivitis (46.1%) and periodontitis (26.9%). These finding are in concordance with the previous studies with reported gingivitis as high as 76%, and irreversible periodontitis reported in 15% of the elite athletes in a 2015 systematic literature review of athletes’ oral health [5]. Recently, Kragt et al. [21] also reported dental plaque accumulation and resultant gingivitis in Olympic athletes assessed by Dutch periodontal screening index of 1.71 ± 0.73. These findings are especially alarming in athletes among whom physical exercise reduces the immune response and increases susceptibility to certain infections, including periodontitis [44,45]. In our study, 21.2% of the athletes presented with erosive tooth wear, which is lower than reported in previous studies that reported presence of erosive tooth wear in 45% [13] and 41.4% [19] of the athletes participating in Olympic Games. The difference may be explained by differing ages of participants. Erosive tooth wear has been shown to increase with age [46] and, compared to the previous studies (mean age > 25 years), our athlete cohort was younger (mean age 18 years).

A large number of athletes also reported that, over the preceding year, oral problems had affected their ability to undertake daily activities (65%) or negatively affected participation and performance in sports (37%). Similarly, several recent studies reported that poor oral health had negative impacts on sport performance in 32–44% of athletes [13,19]. Anecdotal evidence from earlier research (the 1968 and 1992 Olympic Games) revealed a lower prevalence of negative impacts of oral health problems on daily activities (41%) and performance (5%) [47,48]. The observed differences might be due to changes in study methodologies, or enhanced awareness and changes in athletes’ perceptions regarding oral health. Due to time and resource constraints, we used a simplified questionnaire containing only three questions for impact assessment, instead of a detailed and validated questionnaire [49]. As a result, there is possibility that the actual impact of oral problems on daily life and sport participation and performance is much higher than that which we observe in our study.

In the univariate–variate model, the only factor that was able to predict a positive response to questions on the negative impacts of oral health on daily activities/sports participation was periodontal disease (as assessed by BPE). This factor remained significant when the model was adjusted to include demographic factors and the consumption of sports drinks. These findings are in contrast with a recent report by Gallagher et al. [13], who found that self-reported health, but not clinical factors, could predict the impact of oral health on daily activities and participation in sport. However, our findings are not unexpected, as periodontal health has been found to negatively impact health-related quality in the general population [50] and has also been shown to impact athletic performance [13,51]. Our study did not find any significant impact of dental caries and erosive tooth wear on daily activities and sports performance, which might be related to the smaller sample size in the study or differences in clinical assessment or the self-reported impacts.

## 5. Conclusions

This study is the first report on sports and energy drink consumption, oral health status and its impact on the wellbeing and performance of athletes from a developing country. Data was collected using clearly defined outcome measures. Clinical examinations were conducted by a single experienced dentist following standard protocols and guidelines. 

However, the study also has some limitations. First, due to relatively small sample size and recruitment of participants from a single city, the result cannot be generalized for all elite and professional athletes from Pakistan. Secondly, due to the absence of a control group and comprehensive data on oral health of the general population in Pakistan, comparison was not possible. Third, due to financial constraints, radiographs were not taken in this study. Therefore, it is possible that the prevalence of oral diseases was underestimated.

Elite athletes from Pakistan consume sports and energy drinks frequently and have poor oral health, despite self-reported good oral health and hygiene habits. Dental caries, periodontal conditions and erosive tooth wear were common, and periodontal conditions were found to significantly affect ability to undertake daily activities and sports performance. There is an urgent need for oral health screening, awareness, and health promotion strategies in order to minimize oral health problems and their impact on performance among elite athletes from Pakistan.

## Figures and Tables

**Figure 1 nutrients-14-05089-f001:**
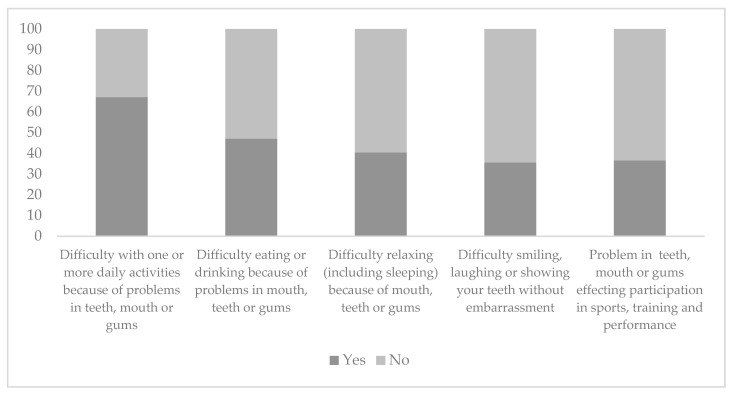
Oral health impact on daily activities and sports participation and performance in athletes (N = 104).

**Table 1 nutrients-14-05089-t001:** Background information of the athletes.

Variables		Frequency (n)	Percent (%)
Gender	Male	84	80.8
	Female	20	19.2
Age	mean ± SD	18.5 ± 4.5	
Ethnicity	Pashtun	91	86.7
	Hindkowi	5	4.8
	Other	7	6.7
Marital status	Single	100	95.2
	Married	4	3.8
Sports Playing	Athletics (track and field)	31	29.8
	Table tennis	39	37.5
	Lawn tennis	15	14.4
	Badminton	6	5.8
	Cricket	6	5.8
	Cycling	2	1.9
	Baseball	2	1.9
	Basket ball	2	1.9
	Hockey	1	1.0
Athletic Caliber	National level	45	43.3
	College/university level	40	38.5
	International	19	18.3
Approximate training (Hours per week)	mean ± SD	26 ± 10.4	
Education	Primary level	14	13.5
	High school	42	40.4
	College/university level	48	46.2
Nutrition education	No	85	81.0
	Yes	19	18.1
Monthly Household Income *	<20,000 PKR	19	18.3
	20,000–50,000 PKR	47	45.2
	>50,000 PKR	8	7.7

***** Missing data.

**Table 2 nutrients-14-05089-t002:** Oral hygiene habits of the athletes.

Variables		Frequency (n)	Percent (%)
Do you brush/clean your teeth regularly?	Yes	102	98.1
	No	2	1.9
How many times do you clean your teeth every day?	One time	51	49.0
	Two or more times	53	51.0
For how long do you clean your teeth?	<1 min	35	33.3
	2 min	43	41.9
	>2 min	26	24.8
When do you clean your teeth? Time	In the morning only	54	51.4
	In the afternoon only	6	5.7
	Before going to bed only	4	3.8
	In the morning and before going to bed	30	28.6
	Other	10	9.5
What do you use for cleaning teeth?	Toothbrush and toothpaste	90	86.5
	Miswak	12	11.5
	Dental floss	2	1.9
Do you use fluoride tooth paste for cleaning teeth?	Yes	36	34.6
	No	25	24.0
	Don’t know	43	41.3
How often do you visit a dentist?	Regularly (after every 6–8 months)	6	5.8
	Occasionally	18	17.3
	Whenever I have dental pain	41	39.4
	I have never visited a dentist	39	37.5
When did you last visit a dentist?	Last 6 months	26	25.0
	Last year	14	13.5
	More than a year	31	29.8
What was the reason for visiting dentist? *	Dental pain	53	50.5
	Family/friend advice	4	3.8
	Doctor/dentist advice	5	4.8
	Other	11	10.5

* Missing data.

**Table 3 nutrients-14-05089-t003:** Sports and energy drink consumption patterns.

Variable	Frequency	Valid Percent
Consumption of energy sports and energy drinks, bars or gels		
Sports and energy drinks	62	59.6
Energy bars	23	22.1
Energy gels	1	1.0
Never consumed any	34	32.7
Brand name(s) commonly consumed		
Sting	33	51.6
Redbull	14	21.9
ORS	9	14.1
Dairy Milk chocolate	10	15.6
Booster	7	10.9
Other	22	34.4
Frequency of consumption (all response data)		
1–2 times per week	33	50.8
3–6 times per week	8	12.3
1 time per day	16	24.6
2 times per day	5	7.7
3 or more times per day	3	4.6
Frequency of consumption (combined data)		
Never or less than 1 time per week	34	34.3
Weekly	41	41.4
Daily	24	24.2
Time of consumption		
Before training	32	47.1
During training	20	29.4
After training	34	50.0
Before competition	3	4.4
During competition	11	16.4
After competition	4	6.0
Reasons for consuming these products		
Provide energy	38	55.9
Replenish lost energy	13	19.4
Replace body fluids	9	13.4
Reduce fatigue	19	28.4
Reduce stress	14	20.9
Improve performance	13	12.4

**Table 4 nutrients-14-05089-t004:** Self-reported oral health status.

Variables		Frequency (n)	Percent (%)
Self-rated General Health	Fair–very poor	14	13.5
	Good–very good	90	86.5
Self-rated Oral Health	Fair–very poor	30	28.8
	Good–very good	74	71.2
Currently or in the past one year, do you have any problem related to your teeth?	At least on problem	83	278.9
	Dental pain	33	31.7
	Sensitivity to hot and cold	27	26.0
	Gum bleeding	9	8.7
	Bad breath	11	10.6
	More than one of these conditions	1	1.0
	Other	2	1.9

**Table 5 nutrients-14-05089-t005:** Crude odds ratio for the association of self-reported oral health, clinical oral health indicators and impact of oral health daily activities on participation in sport.

	Participants (N = 67) Reporting Oral Health Leads to Difficulty with at Least on Daily Activity	Odds Ratio (95% CI)	*p*-Value	Participants (N = 38) Reporting Oral Health Leads t Difficulty with Participation in Sports, Training, and Performance	Odds Ratio (95% CI)	*p*-Value
Self-reported oral health problems						
None	12 (17.9%)	0.679 (0.256–1.803)	0.437	10 (26.3%)	2.390 (0.811–7.043)	0.114
One or more	55 (82.1%)	1		28 (73.7%)	1	
Erosion						
BEWE 0	54 (80.6%)	1.353 (0.509–3.503)	0.557	32 (84.2%)	1.602 (0.547–4.964)	0.390
BEWE one or more	13 (19.4%)	1	0.516	6 (32.0%)	1	
Periodontal conditions						
BPE 0	11 (16.4%)	0.231 (0.093–0.577)	0.002	5 (13.2%)	0.283 (0.097–0.824)	0.021
BPE one or more	56 (83.6%)	1		33 (86.8%)	1	
Dental caries						
None	22 (32.8%)	0.642 (0.281–1.466)	0.293	14 (36.8%)	1.021 (0.446–2.337)	0.961
One or more	45 (67.2%)	1		24 (63.2%)	1	

## Data Availability

Not applicable.

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
