# Peer review of "Sports and Energy Drink Consumption, Oral Health Problems and Performance Impact among Elite Athletes"

_nutrients, 2022, doi:10.3390/nu14235089_

Round 1
Reviewer 1 Report
The authors present an original study on the consumption of energy drinks in young athletes and their relationship with oral health problems and the impact of daily activities and sports performance. The article presents some methodological deficiencies that must be improved in order to be considered for publication:
1- The number of decimals in the text and tables must be homogenized (eg in Table 1 the age appears 18 years and 18.5 in the text).
2- In line 228 it is enough to put % in parentheses, without indicating N, as has been done in the rest of the paper
3- In line 232, it is not clear to which item in table 5 corresponds the 67 (64.4%) mentioned in the text
4- In section 3.6 the authors report that oral health status was not associated with sociodemographic characteristics or oral health and hygiene habits, but what happened to the consumption of sports and energy drinks? Has this possible association been studied? It should be considered, since the consumption of these drinks is the objective of this study
5- In table 6, the categories of the variables add up to 67, not 66 as it appears in the table header. On the other hand, it is not understood where this number comes from, which does not correspond to any of the "Yes" answers in table 5.
6- It is not clear if regression model reported in table 6 is univariate or multivariate, as they say in the discussion. If it is multivariate, the method used should be described in the data analysis section. It is not appropriate to carry out a multivariate model considering as independent variables only those with clinical importance. Multivariate analysis should be preceded by bivariate analysis to determine which variables are associated with response, in addition to including those with clinical importance, as the authors state. They should ensure if they have a sufficient sample size to apply a multivariate model, otherwise they should be limited to descriptive and bivariate analysis.
7- If the dependent variable is impact on athletic performance and impact on daily activities, the possible association between them and the rest of the independent variables included in the study should be studied, not just the four variables shown in table 6.
8- Condition and at least one impact on daily activities an impact on athletic performance, should not be exclusive categories, since a participant can present both. It is suggested that they be treated as two dichotomous variables separately, and the two factor models related to them be studied.
Author Response
Reviewer 1
The authors present an original study on the consumption of energy drinks in young athletes and their relationship with oral health problems and the impact of daily activities and sports performance. The article presents some methodological deficiencies that must be improved in order to be considered for publication:
Response: Thank you for this comment. We have responded to concerns below.
1- The number of decimals in the text and tables must be homogenized (eg in Table 1 the age appears 18 years and 18.5 in the text).
Response: All data is now presented to 1 decimal place.
2- In line 228 it is enough to put % in parentheses, without indicating N, as has been done in the rest of the paper
Response: This has been corrected (N deleted from line 228, line 246 in the revised manuscript) and all data is now presented in a consistent manner
3- In line 232, it is not clear to which item in table 5 corresponds the 67 (64.4%) mentioned in the text
Response: Thank you for picking up on this. To improve clarity, we have changed Table 5 into a clustered column chart (Line 239) and have added data for the item on one or more oral health problem, which relates to the 67 (64.4%). We have also edited the text describing “Oral health impact on daily activities and performance” (sections 3,5) as follows: “Sixty-seven (64.4%) of participants reported difficulty with at least one daily activity because of problems in their mouth, teeth or gums. Nearly half (47.1%) of participants reported that they had difficulty eating or drinking, with 42 (40.4%), 37 (35.5%), and 38 (36.5%) reporting that oral health problems led to difficulty in relaxing, participating in sports, and smiling without embarrassment, respectively (Figure 1)”
4- In section 3.6 the authors report that oral health status was not associated with sociodemographic characteristics or oral health and hygiene habits, but what happened to the consumption of sports and energy drinks? Has this possible association been studied? It should be considered, since the consumption of these drinks is the objective of this study
Response: Thank you for this comment. We have added the following to the methods: “Differences in BEWE, BPE, DMFT were all non-normally distributed, and differences in these measures based on sociodemographics, teeth-brushing habits and reported energy drink consumption (yes/no) was assessed by Mann-Whitney U-test or Kruskal-Wallis test” (Line 191). Further, we have added the following to the results: “A higher proportion of young participants (P=0.011) and a higher proportion of married participants (P=0.005) self-reported at least one oral health problem. Likewise, self-reported oral health problems differed across education status (P=0.005), with those with college/university-level education differing from expected values (adjusted residual -3.1 to 3.1). Mother tongue, sport played, level of education, nutrition education, income, teeth brushing, and consumption of sports/energy drinks, bars and gels did not impact self-reported oral health “(Line 247 - 253), and“ DMFT score was higher in males compared to females (2.0 vs 0.5, P=0.013) and differed across sport played (P=0.039)“ (Line 276 - 278); “BPE differed across sport played (P=0.022) and income categories (P=0.016). BPE was also higher in those consuming sports/energy drinks (1.5 vs 2.0, P=0.037) and was lower in those who stated they never consumed any sports/drinks or bars or gels (1.0 vs 2.0, P=0.008), but no differences were found when sports bars and gels were evaluated independently” (Line 282 - 286); and “BEWE (differed across sports played (P=0.039) and mother tounge (P=0.002) but there were no other statistically significant differences in The normality on demographics and consumption of sports/energy drinks, gels or bars. Likewise, there were no differences for any clinical measures or oral health (DMFT, BPE, BEWE) based on age, marital status, athletic calibre, level of education, and nutrition education. There were also no differences based on teeth-brushing(yes/no) and frequency of teeth brushing(1/day vs 2-3/day), but only two respondents reported they never brushed their teeth” (Line 287 - 294)
5- In table 6, the categories of the variables add up to 67, not 66 as it appears in the table header. On the other hand, it is not understood where this number comes from, which does not correspond to any of the "Yes" answers in table 5.
Response: Due to table 5 being converted into a figure, Table 6 is now Table 5. The 66 was a typographical error and has been corrected to 67. 67 is the number of participants who reported at least one negative impact of oral health on daily activities (this data point has now been added to Figure 1, as mentioned above). 38 is the number of participants who reported that oral health had a negative impact on training (also, as presented in Figure 1 and previously presented in Table 5). To improve clarity, we have edited the titles in Table 5 (previously Table 6) to be (1) Participants (N =68) reporting oral health leads to difficulty with at least on daily activity and (2) Participants (N-38) reporting oral health leads t difficulty with participation in sports, training and performance
6- It is not clear if regression model reported in table 6 is univariate or multivariate, as they say in the discussion. If it is multivariate, the method used should be described in the data analysis section. It is not appropriate to carry out a multivariate model considering as independent variables only those with clinical importance. Multivariate analysis should be preceded by bivariate analysis to determine which variables are associated with response, in addition to including those with clinical importance, as the authors state. They should ensure if they have a sufficient sample size to apply a multivariate model, otherwise they should be limited to descriptive and bivariate analysis.
Response: Thank you for this comment. We have re-run the models as univariate and removed references to multivariate analyses. We have run these as both crude and adjusted models. Not all independent variables were included in the adjusted models, but those found to be statically significant in previous univariate analyses were. Our previous description of regression models has been deleted and replaced with “Clinical indicators of oral health were dichotomized as follows: erosion (BPE 0 vs BPE ≥1), periodontal conditions (BPE 0 vs BPE ≥1), and presence of dental caries (DMFT vs DMFT ≥1); the impact of these factors and self-reported oral health (at least one problem versus no problems) on daily activities (no impact versus at least on daily activity impacted) and participation in sports, training and performance (yes/no impact) was assessed using crude and adjusted logistic regression” (Line 196 - 201). We have also amended the result on regression models as follows: “Adjusted models were run to correct for variables that were found to have a statistically significant impact on BPE (sport played, income, sports drink consumption, sports/energy drink or bar or gel consumption). For impact on daily activities, the adjusted logistic regression model was statistically significant χ2(5) = 14.402, P=0.013. BPE remained significant (0.163 [0.042, 0.628], P=0.008), and was the only significant variable in the model. The model explained 54.1% (Nagelkerke R2) of the variance in participants and correctly classified 47.4% of cases.
For impact on participation in sports, the adjusted logistic regression model was not statistically significant χ2(5) = 9.735, p=0.083 however, BPE remained a significant predictor of impact on participation in sport (0.154[0.029, 0.818], P=0.028)” (Line 308 - 317)
7- If the dependent variable is impact on athletic performance and impact on daily activities, the possible association between them and the rest of the independent variables included in the study should be studied, not just the four variables shown in table 6.
Response: Thank you for this suggestion. In this analysis, we were only interested in WHICH measures of oral health were likely to be contributing to impact on daily activities and therefore, other independent variables were not initially. However, we now included a more detailed assessment of the impact of demographics, and sports drink consumption (i.e. other independent variables) on self-reported oral health, BPE, BEWE and DMFT as per our response to your help comment about section 3.6. As above, we then used any significant factors (gender, sport played, income, sports drink consumption, sports/energy drink or bar or gel consumption) to run an adjusted regression model.
8- Condition and at least one impact on daily activities an impact on athletic performance, should not be exclusive categories, since a participant can present both. It is suggested that they be treated as two dichotomous variables separately, and the two factor models related to them be studied.
Response: Thank you for this comment. We have already treated these as two-factor models. To make this clearer, we have re-structured the regression table – the left side relates to daily activities and the right side to athletic performance.
Regards
Dr. Muhammad Shahzad
Professor (Associate)
School of Biological Sciences, Health and Life Sciences Building
University of Reading, Reading RG6 6AX, United Kingdom

Reviewer 2 Report
This is a very interesting topic and one that has much scope for future studies. I have some points to raise:
i) LINE 74 It is possible inaccurate to say that 'athletes are at high risk of injuries', when the sports they are doing have not been taken into consideration. This statement needs to be qualified. Do the authors mean all athletes, elite athletes, those athletes doing certain contact sports?
ii) LINE 112 The study by Wells & Steele [28] refers to older people and therefore may not be the most appropriate reference to use here. There is no evidence presented here that athletes, a younger cohort, have so many teeth missing that they cannot masticate their food
iii) RESULTS What was the response rate to the questionnaire?
iv) How do the results presented relate to the general population of Pakistan?
v) It would be helpful to know what 'Sting' contains, what the pH is , how much sugar and what type of sugar
vi) Did you ask if the athletes were provided with information about the drinks/gels they consumed? This would be really interesting as it could be significant whether the athletes are told to consume these drinks
vii) The authors need to be clear that the negative effect of poor oral health and sports drinks on performance is a perception; there is no attempt to actually measure/record an effect. This is very difficult to achieve.
viii) It would be useful to know more about the results of the dental screening and relate them to the data collected on the questionnaire. Currently the two sets of data are quite separate.
ix) LINE 281 some aspects of the discussion should be in the results
x) LINE 330 Should read Gallagher et al not Julie et al.
Author Response
Reviewer 2
This is a very interesting topic and one that has much scope for future studies. I have some points to raise:
Response: Thank you for this comment. We have responded to concerns below.
- i) LINE 74 It is possible inaccurate to say that 'athletes are at high risk of injuries', when the sports they are doing have not been taken into consideration. This statement needs to be qualified. Do the authors mean all athletes, elite athletes, those athletes doing certain contact sports?
Response: Thanks for pointing out this. The statement has been corrected in the revised manuscript. It now read as “Since elite athletes especially those in contact sports are at high risk of injuries, the research on oral health in athletes has traditionally limited to oral trauma and use of the mouthguards in athletes” Line 74
- ii) LINE 112 The study by Wells & Steele [28] refers to older people and therefore may not be the most appropriate reference to use here. There is no evidence presented here that athletes, a younger cohort, have so many teeth missing that they cannot masticate their food.
Response: True. Here we meant that missing teeth in athletes, just like older people, may compromise their digestion and energy requirements but the evidence exist only for older people. Since athletes cannot be compared to older people, we deleted the statement and reference in the revised manuscript.
iii) RESULTS What was the response rate to the questionnaire?
Response: The response rate was calculated (83.2%) and presented in the revised manuscript (Line 213).
- iv) How do the results presented relate to the general population of Pakistan?
Response: Thanks for pointing towards this. A discussion point on the same is included in the revised manuscript (Line 316 – 364 and 368 – 376)
- v) It would be helpful to know what 'Sting' contains, what the pH is , how much sugar and what type of sugar.
Response: Thanks for the suggestion. A brief description about the Sting is included in the revised manuscript (Line 351 – 355).
- vi) Did you ask if the athletes were provided with information about the drinks/gels they consumed? This would be really interesting as it could be significant whether the athletes are told to consume these drinks.
Response: Although we asked the athletes about the reasons why they consume the energy drinks, bras or gels, there was no specific question on their source of information e.g. coach, peers or fellow athletes.
vii) The authors need to be clear that the negative effect of poor oral health and sports drinks on performance is a perception; there is no attempt to actually measure/record an effect. This is very difficult to achieve.
Response: Yes, absolutely. The study relied only on self-reported impact, the only possible measure of an impact in the current study circumstances. It will be extremely difficult (if not impossible) to actually measure the impact of oral health problems on sports performance.
Response:
viii) It would be useful to know more about the results of the dental screening and relate them to the data collected on the questionnaire. Currently the two sets of data are quite separate.
Response: Thank you for this comment. We have added the following to the methods: “Differences in BEWE, BPE, DMFT were all non-normally distributed, and differences in these measures based on sociodemographics, teeth-brushing habits and reported energy drink consumption (yes/no) was assessed by Mann-Whitney U-test or Kruskal-Wallis test” (Line 191). Further, we have added the following to the results: “A higher proportion of young participants (P=0.011) and a higher proportion of married participants (P=0.005) self-reported at least one oral health problem. Likewise, self-reported oral health problems differed across education status (P=0.005), with those with college/university-level education differing from expected values (adjusted residual -3.1 to 3.1). Mother tongue, sport played, level of education, nutrition education, income, teeth brushing, and consumption of sports/energy drinks, bars and gels did not impact self-reported oral health “(Line 247 - 253), and“ DMFT score was higher in males compared to females (2.0 vs 0.5, P=0.013) and differed across sport played (P=0.039)“ (Line 276 - 278); “BPE differed across sport played (P=0.022) and income categories (P=0.016). BPE was also higher in those consuming sports/energy drinks (1.5 vs 2.0, P=0.037) and was lower in those who stated they never consumed any sports/drinks or bars or gels (1.0 vs 2.0, P=0.008), but no differences were found when sports bars and gels were evaluated independently” (Line 282 - 286); and “BEWE (differed across sports played (P=0.039) and mother tounge (P=0.002) but there were no other statistically significant differences in The normality on demographics and consumption of sports/energy drinks, gels or bars. Likewise, there were no differences for any clinical measures or oral health (DMFT, BPE, BEWE) based on age, marital status, athletic calibre, level of education, and nutrition education. There were also no differences based on teeth-brushing(yes/no) and frequency of teeth brushing(1/day vs 2-3/day), but only two respondents reported they never brushed their teeth” (Line 287 - 294)
- ix) LINE 281 some aspects of the discussion should be in the results.
Response: Thanks for the suggestion. The whole paragraph has been revised (Line 355 – 362)
- x) LINE 330 Should read Gallagher et al not Julie et al.
Response: The text has been change as suggested (Line 430)
Regards
Dr. Muhammad Shahzad
Professor (Associate)
School of Biological Sciences, Health and Life Sciences Building
University of Reading, Reading RG6 6AX, United Kingdom

Reviewer 3 Report
It is a cross-sectional study. It is well-written. However, it presented some drawbacks. The sample size is small. It focused to a specific region in Pakistan and the results couldn't be generalized.
The authors in my opinion need to include the questionnaire as anex. Also, the authors would describe in more detail which criteria used to calculate the parameter D in DMFT index.
Author Response
Reviewer 3
It is a cross-sectional study. It is well-written. However, it presented some drawbacks. The sample size is small. It focused to a specific region in Pakistan and the results couldn't be generalized.
Response: Thank you for this comment. We agree the results cannot be generalized and have mentioned this in the limitations. Line 443 - 445: “First, due to the relatively smaller sample size and recruitment of participants from a single city, the result cannot be generalized for all elite and professional athletes from Pakistan”. Despite this drawback, we believe the results make an important contribution to the literature because of the lack of published data on the oral health of athletes in Pakistan.
The authors in my opinion need to include the questionnaire as anex. Also, the authors would describe in more detail which criteria used to calculate the parameter D in DMFT index.
Response: Thank you for this comment. We have now provided the questionnaires as supplementary material. Additional detail has been added regarding D has been proved as follows “The presence of dental caries, literally known as tooth decay was assessed using a metallic periodontal probe. The decay was confirmed the presence of carious lesion occlusal or smooth surface, an unmistakable cavity, undermined enamel, or a detectably softened floor or wall”. Line 167 - 170
Regards
Dr. Muhammad Shahzad
Professor (Associate)
School of Biological Sciences, Health and Life Sciences Building
University of Reading, Reading RG6 6AX, United Kingdom

Round 2
Reviewer 2 Report
The changes that have been made following review 1 seem to have improved the manuscript. One limitation is that it can only apply to athletes from Pakistan and cannot be taken as indicative of all athletes.